# MTHFR Polymorphism and Folic Acid Supplementation Influence Serum Homocysteine Levels in Psoriatic Patients Treated with Methotrexate

**DOI:** 10.3390/jcm11154580

**Published:** 2022-08-05

**Authors:** Qi Zhang, Jinran Lin, Zhenghua Zhang, Ling Han, Qiong Huang, Jie Zhu, Bing Wang, Xu Fang, Zhizhong Zheng, Nikhil Yawalkar, Jun Liang, Kexiang Yan

**Affiliations:** 1Department of Dermatology, Huashan Hospital, Fudan University, Shanghai 200040, China; 2Department of Dermatology, Inselspital, Bern University Hospital, University of Bern, 3010 Bern, Switzerland

**Keywords:** psoriasis, methylenetetrahydrofolate reductase, *rs1801133*, methotrexate, homocysteine

## Abstract

Background: Hyperhomocysteinemia has been reported in psoriasis. We investigated the effect of methylenetetrahydrofolate reductase (MTHFR), polymorphism and folic acid supplementation on serum homocysteine levels in psoriasis. Methods: Serum homocysteine levels were detected at baseline and at week 12 in 201 patients who were genotyped with MTHFR rs1801133 without and 93 psoriatic patients with folate supplement. Results: TT genotype carriers of MTHFR rs1801133 had significantly higher serum homocysteine levels at baseline and at week 12, a better PASI 75 response rate at week 8, and a higher PASI 90 response rate at week 12 than the CT and CC genotype carriers. Multiple regression analysis demonstrated that serum homocysteine concentration at baseline was significantly associated with sex, weight, PASI score at baseline, and the rs1801133 genotype. The significant upregulation of serum homocysteine levels after treatment with methotrexate (MTX) was only observed in male CT and CC genotype carriers and female CC genotype carriers. In contrast, folic acid supplementation significantly decreased serum homocysteine levels after MTX treatment but only in male psoriatic patients. Conclusions: The effect of MTX on serum homocysteine levels was associated with the polymorphism of MTHFR rs1801133 and sex. Folic acid supplementation only decreased serum homocysteine levels in male psoriatic patients.

## 1. Introduction

Psoriasis is a chronic inflammatory disease with a worldwide prevalence of 2–3% [1]. Recent studies have demonstrated that patients with psoriasis have significantly higher serum homocysteine levels and a higher prevalence of hyperhomocysteinemia [2,3]. Deficiencies or genetic polymorphisms in the enzymes involved in homocysteine metabolism and insufficient amounts of cofactors are the primary causes of hyperhomocysteinemia [4].

Methotrexate (MTX), a folate derivative that inhibits several enzymes responsible for nucleotide synthesis, has been used as a first-line treatment for moderate-to-severe psoriasis for more than 50 years [5]. Treatment with MTX causes increased serum levels of homocysteine in rheumatoid arthritis (RA), psoriasis, and other diseases [2,6,7,8]. Serum homocysteine levels have been associated with an increased risk of stroke and coronary artery disease [9,10]. However, other studies indicate that MTX decreases the risk of cardiovascular disease [11]. Therefore, the relationship between MTX and serum homocysteine levels is an important indicator for evaluating the efficacy and side effects of MTX in patients with psoriasis.

Methylenetetrahydrofolate reductase (MTHFR), a folate-dependent enzyme, plays an important role in the conversion of the amino acid homocysteine to another amino acid, methionine. MTHFR activity has an adverse effect on serum homocysteine levels [12]. MTHFR *rs1801133* polymorphism consists of a C to T substitution at position 677, which results in an alanine to valine conversion [13]. This missense mutation results in an approximately 70% and 35% reduction in normal MTHFR enzyme activity in TT and CT genotype carriers, respectively [14]. Furthermore, the TT allele is associated with elevated serum homocysteine levels [4]. Whether this genotype affects MTX efficacy and serum homocysteine levels has been the subject of recent debate.

Therefore, this study aims to investigate the effect of MTX on serum homocysteine levels with or without folic acid supplementation and the association of serum homocysteine levels with disease severity, the efficacy of MTX, and single-nucleotide polymorphism (SNP) genotype of the MTHFR *rs1801133* gene.

## 2. Materials and Methods

### 2.1. Patients and Study Design

A total of 294 patients with psoriasis were recruited from our prospective MTX treatment follow-up cohort at the Department of Dermatology, Huashan Hospital, Fudan University between October 2016 and December 2019. All patients received low-dose MTX treatment for 12 weeks, after which their serum homocysteine levels were detected. Patients without serum homocysteine levels determined at baseline and week 12 were excluded. There were 201 psoriatic patients (145 men and 56 women) who did not take folic acid between October 2016 and November 2018. Moreover, we genotyped these patients with MTHFR rs1801133. Folic acid supplementation (24 h after taking MTX) was recommended in 93 (62 men and 31 women) of 294 patients between November 2018 and December 2019. We did not genotype these patients. The medical ethics committee of Huashan Hospital approved the protocol (approval MTX201501), and all patients provided written, informed consent. Patients aged ≥ 18 years were recruited from the outpatient population. Only patients who had moderate-to-severe psoriasis with or without arthritis were included in this study. Patients receiving ultraviolet radiation therapy, MTX, or other systemic treatments for psoriasis or arthropathy within one month of the study initiation were excluded. Topical treatments were stopped two weeks before the start of MTX treatment. The European guidelines on the contraindications for and restrictions on the use of MTX were followed.

### 2.2. Treatment Strategy and the Evaluation of Serum Homocysteine Level

The initial MTX dosage was 7.5–10 mg orally once weekly. The dosage was increased by 2.5 mg every 2–4 weeks to a maximum of 15 mg weekly depending on clinical response, side effects, and the results of routine hematology and chemistry tests. If liver enzyme elevations were greater than 2-fold but less than 3-fold, the MTX dosage was reduced by 2.5 mg weekly and administered 2–4 weeks later. If liver enzyme elevations exceeded 3-fold, MTX treatment was stopped.

Two certified dermatologists graded psoriasis using the PASI and body surface area (BSA) scores at baseline and at weeks 4, 8, and 12. Lipid profiles, serum homocysteine levels at baseline and at 12 weeks for MTX treatment, and fasting blood glucose at baseline were measured using conventional laboratory techniques at Huashan Hospital. Sex, age, age at disease onset, smoking, alcohol intake, hypertension, diabetes, height, weight, and body mass index (BMI) were recorded.

### 2.3. DNA Extraction and Genotyping Analysis

Five milliliters of EDTA-anticoagulated whole blood was collected from all patients and stored at −80 °C. Genomic DNA was extracted from peripheral blood lymphocytes using the FlexiGene DNA Purification Kit (Qiagen, Hiden, Germany) and diluted to 20 ng/µL. All DNA samples were stored at −20 °C. The *rs1801133* SNP of the MTHFR gene was genotyped using a SequenomMassARRAY.

The SequenomMassARRAY Assay Design 3.0 software was used to design the PCR parameters and detection primers. The PCR products were subsequently used as templates for locus-specific, single-base extension reactions. The resulting products were desalted and transferred to a 384-element SpectroCHIP array (Sequenom Inc., San Diego, CA, USA). MALDI-TOF MS (Sequenom Inc.) was used for allele detection. The mass spectrograms were analyzed using MassARRAY Type software (Sequenom Inc.). We performed quality control of SNPs and samples at a call rate of 99.7% and analyzed the distribution at *rs1801133* in the HCs with the Hardy–Weinberg equilibrium (*p* > 0.0001).

### 2.4. Statistical Analysis

Descriptive statistics were presented as mean and SD for continuous measures and frequencies and percentages for categorical measures. Statistical analyses were performed using a one-way analysis of variance (Newman–Keuls), chi-square test, unpaired t-test, paired t-test, and Wilcoxon signed-rank test as appropriate. Multivariate analyses were performed using the statistical software package SPSS 23.0 (IBM Corp, Armonk, NY, USA) after adjusting for age, sex, disease duration, disease severity by PASI and BSA scores, *rs1801133* genotype, smoking, alcohol consumption, hypertension, and diabetes. Results with *p* < 0.05 were considered statistically significant.

## 3. Results

### 3.1. TT Genotype of MTHFR rs1801133 Was Associated with Higher Serum Homocysteine Level and Better Response to MTX

As shown in Table 1, TT and CT genotype carriers at *rs1801133* had a younger age than CC genotype carriers (33.3 ± 16.2 vs. 31.8 ± 15.0 vs. 37.7 ± 15.7, *p* = 0.0149). Moreover, the levels of serum homocysteine at baseline (21.58 ± 13.78 vs. 13.57 ± 3.74 vs. 13.15 ± 4.21, *p* < 0.0001) and at week 12 (21.7 ± 12.14 vs. 14.98 ± 5.16 vs. 14.93 ± 5.78, *p* < 0.0001), the PASI 75 response rate at week 8 (42.9% vs. 23.9% vs. 21.6%, *p* = 0.0487), and the PASI 90 response rate at week 12 (40.0% vs. 21.7% vs. 16.2%, *p* = 0.0208) were significantly higher in the TT genotype carriers at *rs1801133* than the TC and CC genotypes carriers. No significant differences were observed in other characteristics.

### 3.2. Serum Homocysteine Level Was Associated with the MTHFR Polymorphism, Sex, MTX Efficacy, and Disease Severity in Psoriasis

As shown in Table 2, univariate regression analysis demonstrated sex (*p* = 0.001), weight (*p* = 0.024), PASI score at baseline (*p* = 0.007), the genotype at *rs1801133* (*p* = 0.000), serum levels of ApoA1 (*p* = 0.008), and HDL-C (*p* = 0.012) were statistically associated with serum homocysteine levels at baseline. However, only sex (*p* = 0.004), weight (*p* = 0.04), PASI score at baseline (*p* = 0.038), and *rs1801133* genotype (*p* = 0.000) significantly correlated with serum homocysteine levels at baseline in multiple regression analysis after adjustments were made for these factors. In addition, serum homocysteine levels at week 12 after MTX treatment without folic acid supplementation was significantly associated with sex (*p* = 0.000), mean PASI improvement at week 12 (*p* = 0.009), *rs1801133* genotype (*p* = 0.000)*,* and PASI score at baseline (*p* = 0.022) in univariate regression analysis. However, only sex (*p* = 0.000), mean PASI improvement at week 12 (*p* = 0.006), and *rs1801133* genotype (*p* = 0.000) significantly correlated with serum homocysteine levels at week 12 in multiple regression analysis after adjustments were made for these factors.

### 3.3. The Effect of MTX on Serum Homocysteine Level Was Related to Sex and the Polymorphism of MTHRF rs1801133 without Folic Acid Supplementation

Table 3 summarizes the effect of MTX on serum homocysteine levels according to sex and the *rs1801133* genotype. MTX significantly upregulated serum homocysteine levels in the CT (13.57 ± 3.74 vs. 14.98 ± 5.16, *p* = 0.0004) and CC (13.15 ± 4.21 vs. 14.93 ± 5.78, *p* < 0.0001) *rs1801133* genotype carriers but not in TT (21.58 ± 13.78 vs. 21.7 ± 12.14, *p* > 0.05) genotype carriers without folate supplementation. Moreover, a significant effect of MTX on serum homocysteine level was only observed in male CT (14.41 ± 3.89 vs. 16.14 ± 5.39, *p* = 0.0011) and CC (14.17 ± 4.02 vs. 16.24 ± 5.91, *p* = 0.0005) genotype carriers and female CC (10.19 ± 3.30 vs. 11.14 ± 3.19, *p* = 0.0081) genotype carriers.

### 3.4. Folic Acid Supplementation Reverses the Upregulation of MTX on Serum Homocysteine Levels in Male Psoriatic Patients

We further analyzed the effect of folic acid supplementation on serum homocysteine levels during MTX treatment (Table 4). Our results show that MTX significantly downregulated serum homocysteine levels with folic acid supplementation (14.76 ± 11.7 vs. 13.44 ± 7.52, *p* = 0.0012), which was opposite to the upregulation of MTX in serum homocysteine levels without folic acid supplementation (14.81 ± 7.41 vs. 16.13 ± 7.48, *p* = 0.001). However, the decrease in serum homocysteine after folic acid supplementation was only observed in male psoriatic patients (16.84 ± 13.74 vs. 14.6 ± 8.34, *p* = 0.0002), not in their female counterparts (10.61 ± 3.04 vs. 11.13 ± 4.86, *p* > 0.05).

## 4. Discussion

In this study, we found that serum homocysteine levels were related to the MTHFR *rs1801133* genotype and the PASI score of psoriasis patients. MTX treatment further elevated serum homocysteine levels, and folate supplementation reversed this increase among male psoriasis patients.

Pharmacogenomics explores the relationship between individual genetic variants and drug effects to predict efficacy and toxicity, for example, MTX in patients with psoriasis [15]. Our previous study demonstrated that the TT genotype at *rs10036748* in TNIP1 is associated with the PASI 75 and PASI 90 response to MTX in psoriasis patients, whichis related to MTX-inhibited NF-κB activation, and the variants of TNIP1 might correlate with the activation of NF-κB [16]. The role of TNIP1 on serum homocysteine levels has so far not been determined and remains to be elucidated in future studies. In this study, we found that MTHFR *rs1801133* TT genotype carriers had significantly higher serum homocysteine levels and a better therapeutic efficacy of MTX than the CT and CC genotype carriers. These results suggest that the increased serum homocysteine levels might not be a contraindication of MTX for psoriasis patients. However, we still recommend monitoring serum homocysteine levels and side effects, as a meta-analysis showed that the TT *rs1801133* genotype had a 1.84-fold increased risk of hemorrhagic stroke compared with the CC or CT genotypes [17]. In the present study, we found that patients with the TT genotype showed significantly higher serum homocysteine levels before starting MTX therapy than patients with the wild-type (CC) or CT genotype. Nonetheless, after MTX therapy, serum homocysteine levels were significantly increased among male patients with the CC and CT genotypes and female patients with the CC genotype, but neither in male nor female TT genotype patients. These results were not consistent with other studies that showed that MTHFR *rs1801133* polymorphism modifies the toxicity of MTX and indicates that patients with the TT genotype are at increased risk of developing hyperhomocysteinemia after MTX treatment in patients with ovarian cancer [18]. This diversity might be due to the different underlying diseases.

This study found that primary serum homocysteine levels in male psoriasis patients were higher than in female patients. An analysis of healthy people demonstrated that serum homocysteine levels are higher in males than in females [19]. Elevated serum homocysteine levels have been reported in male patients with hypertension, cardiovascular disease, primary chronic venous disease, and glaucomatous disease [20,21,22,23]. Previous studies also indicate that further factors, such as a surge in blood lead and cadmium levels, could increase homocysteine levels in males [24]. In addition, some factors expressed differently in men and women, such as creatine synthesis and estrogen levels, might also contribute to higher serum homocysteine levels in male psoriasis patients [25,26].

Folate, a form of vitamin B9, aids the performance of many functions, including cell division, growth, and the production of new red blood cells [27]. A previous clinical study reported a protective effect of folic acid supplementation for patients with rheumatoid arthritis during treatment with MTX [28]. Our results support that folate could down-regulate serum homocysteine levels in MTX-treated psoriasis patients; however, this effect was sex-dependent as folate downregulated serum homocysteine levels in male patients only. This finding suggests we need to consider sex when giving folic acid supplements to patients with psoriasis who receive MTX treatment.

Homocysteine is affected by multiple factors, including the patient’s genetic background and lifestyle. Yang et al. conducted a population-based cross-sectional study, including a total of 4012 participants, and found that a higher BMI and older age were potential risk factors for hyperhomocysteinemia. They also found that current smoking was associated with a higher risk of hyperhomocysteinemia. Furthermore, alcohol consumption or education level might also influence the risk of hyperhomocysteinemia, whereas fruit and vegetable consumption may have a protective effect against hyperhomocysteinemia [29]. Our study also found a correlation between body weight and homocysteine. However, we did not find a correlation between smoking, alcohol consumption or age and homocysteine levels (data not shown), indicating that these factors did not appear to be significant in raising homocysteine in our cohort. We did not analyze the education level or fruit and vegetable consumption of the patients in this study.

In conclusion, this study demonstrates that the level of serum homocysteine might not only positively correlate psoriasis severity but also with the curative effect of MTX. Based on these findings, we believe that MTX is safe and effective for treating psoriasis patients under close monitoring, while folic acid supplementation is beneficial for male patients.

## Figures and Tables

**Table 1 jcm-11-04580-t001:** The difference between clinical characteristics and serum homocysteine levels according to the genotype of MTHFR *rs1801133* in 201 psoriatic patients.

	*rs1801133* (*n* = 201)			
	TT (*n* = 35)	CT (*n* = 92)	CC (*n* = 74)	*p*-Value
Male [*n*(%)]	24 (81.7)	66 (71.7)	55 (74.3)	0.8168
Age (years), mean ± SD	47.0 ± 14.9	44.4 ± 16.0	49.8 ± 13.6	0.0731
Age at disease onset, mean ± SD	33.3 ± 16.2	31.8 ± 15.0	37.7 ± 15.7	0.0149
Disease duration, mean ± SD	13.7 ± 10.6	12.7 ± 10.5	12.3 ± 9.2	0.7951
Body mass index (kgm^−2^), mean ± SD	24.3 ± 4.0	24.2 ± 3.35	25.1 ± 3.4	0.2547
PASI at baseline, mean ± SD	15.5 ± 6.0	13.8 ± 7.1	13.9 ± 8.0	0.4942
BSA (%) at baseline, mean ± SD	29.4 ± 17.9	28.1 ± 21.7	26.4 ± 20.4	0.7467
Smoking [*n*(%)]	16 (45.7)	24 (26.1)	23 (31.1)	0.1032
Drinking alcohol [*n*(%)]	9 (25.7)	22 (23.9)	22 (29.7)	0.6963
Hypertension [*n*(%)]	14 (40.0)	27 (29.3)	26 (35.1)	0.4805
Diabetes [*n*(%)]	8 (22.9)	15 (16.3)	11 (14.9)	0.5698
Arthritis [*n*(%)]	15 (42.9)	42 (45.7)	47 (63.5)	0.2783
Complains of side effects [*n*(%)]	6 (17.1)	32 (34.8)	25 (33.8)	0.136
MTX dosage(mg), mean ± SD	130.1 ± 16.4	138.8 ± 22.5	136.5 ± 18.1	0.098
Serum homocysteine level at baseline (μmol/L)	21.58 ± 13.78	13.57 ± 3.74	13.15 ± 4.21	<0.0001
Serum homocysteine level at week 12 (μmol/L)	21.7 ± 12.14	14.98 ± 5.16	14.93 ± 5.78	<0.0001
Outcomes at week 8				
PASI50	23 (65.7)	54 (58.7)	39 (52.7)	0.4241
PASI75	15 (42.9)	22 (23.9)	16 (21.6)	0.0487
PASI90	4 (11.4)	7 (7.6)	7 (9.5)	0.7826
The mean PASI improvement	55.1 ± 40.58	51.5 ± 29.5	48.2 ± 33.3	0.5855
Outcomes at week 12				
PASI50	28 (80.0)	69 (75.0)	48 (64.9)	0.1828
PASI75	22 (62.9)	46 (50.0)	29 (39.2)	0.0628
PASI90	14 (40.0)	20 (21.7)	12 (16.2)	0.0208
The mean PASI improvement	68.1 ± 41.3	64.9 ± 29.1	61.3 ± 30.4	0.5593

Abbreviations: BSA, body mass area; MTX, methotrexate; PASI, Psoriasis Area Severity Index. PASI50, 50% reduction from baseline PASI score; PASI75, 75% reduction from baseline PASI score; PASI90, 90% reduction from baseline PASI score. Chi-square or one-way analysis of variance (Newman–Keuls) were used as appropriate. The results with *p* < 0.05 were considered statistically significant.

**Table 2 jcm-11-04580-t002:** The factors associated with serum homocysteine level at baseline and week 12 in univariate analysis and multiple analysis.

	Predictors	Univariate Analysis		Multiple Analysis	
		OR (95%CI)	*p*-Value	OR (95%CI)	*p*-Value
Homocysteine at baseline	sex	−3.997 (−6.232~−1.762)	0.001	−3.385 (−5.682~−1.088)	0.004
	weight	0.092 (0.013–0.171)	0.024	0.081 (0.004–0.158)	0.040
	PASI score at baseline	0.195 (0.055–0.334)	0.007	0.142 (0.008–0.276)	0.038
	*rs1801133*	−3.546 (−4.913~−2.179)	0.000	−3.713 (−5.083~−2.342)	0.000
	ApoA1	−8.464 (−14.675~−2.253)	0.008		
	HDL-C	−4.887 (−8.682~−1.092)	0.012		
Homocysteine at week 12	sex	−5.008 (−7.226~−2.791)	0.000	−5.339 (−7.432~−3.245)	0.000
	the mean PASI improvement at week 12	4.301 (1.08–7.522)	0.009	4.156 (1.200–7.112)	0.006
	*rs1801133*	−2.799 (−4.213~−1.384)	0.000	−2.807 (−4.131~−1.483)	0.000
	PASI score at baseline	0.165 (0.024–0.307)	0.022		

Abbreviation: ApoA1, apolipoprotein A1; HDL-C, high-density lipoprotein cholesterol; PASI, Psoriasis Area Severity Index. Multiple regression analysis was performed after adjustment for sex, weight, PASI score at baseline, ApoA1, HDL-C, the mean PASI improvement at week 12, *rs1801133* genotype.

**Table 3 jcm-11-04580-t003:** The effect of MTX on serum homocysteine levels according to the genotype of MTHFR *rs1801133* and gender.

	Total (*n* = 201)				Male (*n* = 145)				Female (*n* = 56)		
Genotype	0 W	12 W	*p*-Value	Genotype	0 W	12 W	*p*-Value	Genotype	0 W	12 W	*p*-Value
TT (*n* = 35)	21.58 ± 13.78	21.7 ± 12.14	0.9465	TT (*n* = 24)	24.09 ± 14.20	24.29 ± 13.62	0.935	TT (*n* = 11)	16.10 ± 11.58	16.06 ± 4.80	0.083
CT (*n* = 92)	13.57 ± 3.74	14.98 ± 5.16	0.0004	CT (*n* = 66)	14.41 ± 3.89	16.14 ± 5.39	0.0011	CT (*n* = 26)	11.43 ± 2.24	12.02 ± 2.94	0.112
CC (*n* = 74)	13.15 ± 4.21	14.93 ± 5.78	<0.0001	CC (*n* = 55)	14.17 ± 4.02	16.24 ± 5.91	0.0005	CC (*n* = 19)	10.19 ± 3.30	11.14 ± 3.19	0.0081

Paired *t*-test or Wilcoxon matched-pairs signed rank test was used as appropriate. The results with *p* < 0.05 were considered statistically significant.

**Table 4 jcm-11-04580-t004:** The effect of MTX on serum homocysteine levels according to gender and folate supplements.

	Total (*n* = 294)			Male (*n* = 207)			Female (*n* = 87)		
	0 W	12 W	*p*-Value	0 W	12 W	*p*-Value	0 W	12 W	*p*-Value
without folate (*n* = 201)	14.81 ± 7.41	16.13 ± 7.48	0.001	15.92 ± 7.64	17.52 ± 8.06	0.0019	11.93 ± 5.90	12.52 ± 3.84	0.0002
with folate (*n* = 93)	14.76 ± 11.7	13.44 ± 7.52	0.0012	16.84 ± 13.74	14.6 ± 8.34	0.0002	10.61 ± 3.04	11.13 ± 4.86	0.3678

Paired *t*-test or Wilcoxon matched-pairs signed rank test were used as appropriate. The results with *p* < 0.05 were considered statistically significant.

## Data Availability

The datasets presented in this study can be found in online repositories. The names of the repository/repositories and accession number(s) can be found below: EBI EVA, accession no: PRJEB51117.

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
