# Peer review of "MTHFR Polymorphism and Folic Acid Supplementation Influence Serum Homocysteine Levels in Psoriatic Patients Treated with Methotrexate"

_jcm, 2022, doi:10.3390/jcm11154580_

Round 1
Reviewer 1 Report
Dear Authors,
Thank. yoou for the possibility to read such an interesting, prospective study. I have some questions
1. Please adjust the article to JCM authors guidelines - e.g. abstract
2. Why you do not administer MTX in almost 50% of your population. What about the side effects? Is it ethical? Was it a part of the study?
3. I do not understand the clinical usefulness of measuring levels of homocysteine in general (if I administer MTX I will do it regardless of low homocysteine). Moreover in the era of biologics, where MTX is less and less frequently used it is even harder for me to understand .
4. You say that this is a prospective group, however it is 2022 and the last result is from 3 years ago. Please elaborate on that
Author Response
- Please adjust the article to JCM authors guidelines - e.g. abstract
Answer: Thank you for your reminder. We have revised the abstract according to the JCM authors guidelines. We shortened the abstract to less than 200 words as shown as follows:
Background: Hyperhomocysteinemia has been reported in psoriasis. We investigated the effect of methylenetetrahydrofolate reductase (MTHFR) polymorphism and folic acid supplementation on serum homocysteine levels e in psoriasis.
Methods: Serum homocysteine levels were detected at baseline and at week 12 in -201 patients who were genotyped with MTHFR rs1801133 without and 93 psoriatic patients with folate supplement.
Results: TT genotype carriers of MTHFR rs1801133 had significantly higher serum homocysteine levels at baseline and week 12, better PASI 75 response rate at week 8, and a higher PASI 90 response at week 12 than the CT and CC genotype carriers. Multiple regression analysis demonstrated that serum homocysteine concentration at baseline was significantly associated with sex, weight, PASI score at baseline, and the rs1801133 genotype. Significant upregulation of serum homocysteine level after treatment with methotrexat (MTX) was only observed in male CT and CC genotype carriers and female CC genotype carriers. In contrast, folic acid supplementation significantly decreased serum homocysteine levels after MTX treatment but only in male psoriatic patients.
Conclusions: The effect of MTX on serum homocysteine level was associated with the polymorphism of MTHFR rs1801133 and sex. Folic acid supplementation only decreased serum homocysteine levels in male psoriatic patients.
- Why you do not administer MTX in almost 50% of your population. What about the side effects? Is it ethical? Was it a part of the study?
Answer:
Thank you for your question. In my psoriasis clinic, more than 70% patients
received MTX treatment. The side effects that we observed were including:
gastrointestinal tract symptoms such as nausea and vomiting, dyspepsia, abdominal
pain, diarrhea, oral ulcers, fatigue, dizziness, headache, hair loss, elevated hepatic
enzyme, cytopenia, and others. MTX is ethical medicine. The study was reviewed by the ethics committee of Huashan Hospital Fudan University (MTX201501).
This article is a part of our study. We have observed the effect of MTX on lipid profiles and blood viscosity, the relevant articles have been published as:
- Wang B, Deng H, Hu Y, Han L, Huang Q, Fang X, Yang K, Wu S, Zheng Z, Yawalkar N, Zhang Z, Yan K, The difference of lipid profiles between psoriasis with arthritis and psoriasis without arthritis and sex specific downregulation of methotrexate on the apolipoprotein B/apolipoprotein A-1 ratio. Arthritis research & therapy 2022;24: 17.
- Han L, Guo M, Wang B, Meng Q, Zhu J, Huang Q, Zhang Z, Fang X, Yang K, Wu S, Zheng Z, Yawalkar N, Deng H, Yan K, Sex-differential downregulation of methotrexate on plasma viscosity and whole blood viscosity in psoriasis. Clinical hemorheology and microcirculation 2022 Apr 19. doi: 10.3233.
- I do not understand the clinical usefulness of measuring levels of homocysteine in general (if I administer MTX I will do it regardless of low homocysteine). Moreover in the era of biologics, where MTX is less and less frequently used it is even harder for me to understand.
Answer: Thank you for your question. Methotrexate is a traditional and commonly used treatment for psoriasis patient. Because it is effective and cheap, MTX is accepted by most patients in our hospital. Since psoriasis patients have cardiovascular disease risk elevated, detecting serum homocysteine is a very simple method, so we used monitoring the level of Hcy to predict the efficacy and the patient's physical condition when psoriatic patients treated with MTX.
In the era of biologics, we also used biologics in our clinic, we found that biological agents did not reduce the level of atherosclerotic blood lipids in psoriasis patients, and even increased the risk, but MTX can limit the elevation of atherosclerotic blood lipids. Interestingly, MTX down-regulated the levels of inflammatory indexes (plasma blood viscosity, ESR, hCPR) and cardiovascular risk factors ApoB/ApoA1 with sex differences, and down-regulated the levels of these indexes only in male patients. In this study, it was found that folic acid supplementation down-regulated the levels of serum homocysteine only in male patients. This suggests that there may be sex differences in the reduction of cardiovascular disease in patients with psoriasis by MTX, which may also be the reason why some studies did not find that MTX reduced cardiovascular risk. In addition, long-term use of biological agents can cause immunological drift, the occurrence of allergic reactions, and the generation of anti-drug antibodies, thus reducing the efficacy of biological agents. MTX can inhibit the immune response, reduce serum IgE level and limit the production of anti-drug antibodies. Therefore, the combination of MTX and biological agents improves the efficacy and onset speed, reduces the side effects of biologics, and improves the retention rate. We think that combining biologics with MTX may be the best efficacious method in the future. It will be cheaper, and the efficacy will be faster.
- You say that this is a prospective group, however it is 2022 and the last result is from 3 years ago. Please elaborate on that
Answer: Thank you for your question. Actually, we have established nearly 700 psoriatic patients cohort treated with methotrexate until today. We analysed data and wrote this paper in 2019. This year, we found that adding data to the study did not change the results when we re-analyzed the data with more patients with folic acid treatment. So we did not change our data in this manuscript.
Reviewer 2 Report
In my opinion the article is well written and interesting. For many years it is well known that psoriatis patients have cardiovascular disease risk elevated. Nowadays we know it is a problem of inflamation, but also of other factors connected with stress (because of disease), smoking, overweight and so on. Homocysteine, for some years, is seem to be a predicor of cardiovascular diseases. It is very important to recognize psoriatic patients, that we have to be specially carefull of other diseases. However gene polymorfism is not now an everyday blood test, it may help us in the futere to find the best drug for psoratic patients.Comments to the Author
- Interesting topic
- Relevant data in future clinical practice
- Concise written
- The group is quite big, but for 3 years and 2 Departments of Dermatology, the group could be bigger
- It would be interesting to check in the group of the same patients the effect of folic acid suplementation
Author Response
The group is quite big, but for 3 years and 2 Departments of Dermatology, the group could be bigger
Answer: Thank you for your comment. We agree with your opinion. Until this year, we have established nearly 700 psoriatic patients treated with methotrexate and we will continue to do this work.
- It would be interesting to check in the group of the same patients the effect of folic acid suplementation
Answer: Thank you for your suggestion. We agree with your opinion. We will analyze the effect of folic acid in the same patients when we have enough samples. Some patients might stop methotrexate treatment because of severe side effects.
Round 2
Reviewer 1 Report
Dear Authors,
Thank you for answering my questions.
I am sorry about question number 2 - I was thinking about foliate and wrote MTX. Why you do not administer folic acid in 50% of the cases. The rest of the question still stands.
Please elaborate.
Author Response
I am sorry about question number 2 - I was thinking about foliate and wrote MTX. Why you do not administer folic acid in 50% of the cases. The rest of the question still stands.
Answer: Thank you for your question. Actually, we did not supplement patients with folic acid until November 2018, since there was no folic acid in the pharmacy of our hospital before. It might be a pity for these patients. In 2018, when our paper “Safety and Efficacy of Methotrexate for Chinese Adults With Psoriasis With and Without Psoriatic Arthritis” was published in JAMA Dermatology, the reviewers suggested that folic acid should be given to the patients. We agreed with the experts, so we applied to the pharmacy director to introduce folic acid into the hospital pharmacy. From November 2018 to the present, all patients treated with methotrexate are taking folic acid.